# Familial Risks and Proportions Describing Population Landscape of Familial Cancer

**DOI:** 10.3390/cancers13174385

**Published:** 2021-08-30

**Authors:** Kari Hemminki, Kristina Sundquist, Jan Sundquist, Asta Försti, Akseli Hemminki, Xinjun Li

**Affiliations:** 1Faculty of Medicine and Biomedical Center in Pilsen, Charles University in Prague, 30605 Pilsen, Czech Republic; 2Division of Cancer Epidemiology, German Cancer Research Center (DKFZ), Im Neuenheimer Feld 580, 69120 Heidelberg, Germany; 3Center for Primary Health Care Research, Lund University, 20502 Malmö, Sweden; kristina.sundquist@med.lu.se (K.S.); jan.sundquist@med.lu.se (J.S.); a.foersti@dkfz.de (A.F.); xinjun.li@med.lu.se (X.L.); 4Department of Family Medicine and Community Health, Department of Population Health Science and Policy, Icahn School of Medicine at Mount Sinai, New York, NY 10029, USA; 5Center for Community-Based Healthcare Research and Education (CoHRE), Department of Functional Pathology, School of Medicine, Shimane University, Izumo 693-8501, Shimane, Japan; 6Hopp Children’s Cancer Center (KiTZ), 69120 Heidelberg, Germany; 7Division of Pediatric Neurooncology, German Cancer Research Center (DKFZ), German Cancer Consortium (DKTK), 69120 Heidelberg, Germany; 8Cancer Gene Therapy Group, Translational Immunology Research Program, University of Helsinki, 00290 Helsinki, Finland; akseli.hemminki@helsinki.fi; 9Comprehensive Cancer Center, Helsinki University Hospital, 00290 Helsinki, Finland

**Keywords:** familial risk, familial proportion, high-risk families, nationwide study, family-cancer database

## Abstract

**Simple Summary:**

Familial cancer can be defined through the occurrence of the same cancer in two or more family members. Hereditary cancer is a narrower definition of high-risk familial aggregation through identified predisposing genes. The absence of correlation between spouses for risk of most cancers, particularly those not related to tobacco smoking or solar exposure, suggests that familial cancers are mainly due to genetic causes. The aim of the present study was to define the frequency and increased risk for familial cancer. Data on 31 of the most common cancers were obtained from the Swedish Family-Cancer Database and familial relative risks (SIRs) were estimated between persons with or without family history of the same cancer in first-degree relatives. Practically all cancers showed a familial risk, with an SIR most commonly around two, or a doubling of the risk because of family history.

**Abstract:**

Background: Familial cancer can be defined through the occurrence of the same cancer in two or more family members. We describe a nationwide landscape of familial cancer, including its frequency and the risk that it conveys, by using the largest family database in the world with complete family structures and medically confirmed cancers. Patients/methods: We employed standardized incidence ratios (SIRs) to estimate familial risks for concordant cancer among first-degree relatives using the Swedish Cancer Registry from years 1958 through 2016. Results: Cancer risks in a 20–84 year old population conferred by affected parents or siblings were about two-fold compared to the risk for individuals with unaffected relatives. For small intestinal, testicular, thyroid and bone cancers and Hodgkin disease, risks were higher, five-to-eight-fold. Novel familial associations included adult bone, lip, pharyngeal, and connective tissue cancers. Familial cancers were found in 13.2% of families with cancer; for prostate cancer, the proportion was 26.4%. High-risk families accounted for 6.6% of all cancer families. Discussion/Conclusion: High-risk family history should be exceedingly considered for management, including targeted genetic testing. For the major proportion of familial clustering, where genetic testing may not be feasible, medical and behavioral intervention should be indicated for the patient and their family members, including screening recommendations and avoidance of carcinogenic exposure.

## 1. Introduction

Familial cancer can be defined through the occurrence of the same cancer in two or more family members. Hereditary cancer is a narrower definition of high-risk familial aggregation through identified predisposing genes or Mendelian type of inheritance. The absence of correlation between spouses for risk of most cancers, particularly those not related to tobacco smoking or solar exposure, suggests that familial cancers are mainly due to genetic causes [1,2]. Hereditary cancer has become an important issue in oncology clinics because of the success in implementing genetic testing and/or screening methods for cancer syndromes [3,4]. For the management of hereditary cancers, it is essential to identify individuals and families at risk and provide targeted surveillance and management for affected individuals and their family members [4]. High-risk individuals may be identified based on family history and patient-specific personal factors, such as age at diagnosis and tumor phenotype [3,4,5,6]. While ascertainment of family history is still important in most management recommendations, panel sequencing has been brought forward as an early diagnostic tool [7]. Multigene panels allow for the inclusion of a battery of susceptibility genes, which is important because emerging evidence suggests a wider distribution of pathogenic mutations than previously thought. Under-recognition of clinical criteria used to identify individuals with hereditary cancer may lead to incomplete risk assessments and insufficient surveillance recommendations [3].

Risk for familial cancer is a continuum from rare high-risk to common low-risk clustering [2,8,9]. While the need for action regarding high-risk families is generally accepted and management procedures are in practice, there are no consensus guidelines for the much larger group of families with lower risk lacking a dissected genetic background. This is a dilemma at the time when genetic testing is daily news and public awareness of familial cancer and demand for counseling are increasing. For the oncology community, the dilemma culminates in lacking operational guidelines, on how to obtain and judge the family history of various cancers, and if or how to counsel family members. Even though the guidance from professional organizations on diagnostics and management of low-risk familial cancer is lacking, the guidelines for population screening recommendations do consider family histories of breast, colorectal, and prostate cancers and earlier starting ages have been recommended [10,11,12,13]. We have used age-specific risks for familial cancer to empirically define the antedated starting age for population screening of breast and prostate cancers [14,15].

In the present article, we aim to describe a nationwide landscape of concordant familial cancer by defining the relative risk depending on the affected proband (parent or sibling), number of affected probands and, for common cancers, by exact diagnostic age of familial cases. In addition, we will be able to define familial proportions for each cancer thus indicating the share of familial cancers among all cancers. The results are discussed in terms of germline genetic background and clinical implication. The analyses are based on the latest version of the Swedish Family-Cancer Database (FCD), which is a unique source for family studies comprising 16.8 million people with clinical cancer data and other detailed personal information from 1958 through 2016.

## 2. Results

### 2.1. Case Numbers, Median Ages, Incidence, and Familial Proportions

Table 1 lists background data for 9.3 million index individuals and for their 31 cancer sites (colorectal cancer not counted as colon and rectal cancers are separately included) analyzed through the years 1958 to 2016. The ICD-7 codes specify the definitions of each cancer. Overall, 525,019 cancers were diagnosed in the second generation at ages between 20 and 84 years. Breast cancer accounted for 18.2% (crude incidence 30.85/100,000) and prostate cancer 17.5% (57.82/100,000) of all patients. The median age at diagnosis for these second generation patients was 60 years, which is about 10 years lower than that for all cancers in the Swedish Cancer Registry. The median diagnostic age was highest for squamous cell skin (67 years) and prostate cancers (66 years) and lowest for Hodgkin disease (32 years) and testicular cancer (33 years). A total of 69,104 patients had concordant cancers, accounting for 13.2% of all cancers in the offspring generation. More than 1/3 of these were prostate cancers, for which the familial proportion of 26.4% was highest. This was followed by breast (17.5%), colorectal (15.7%), and lung (13.0%) cancers. For rare cancers, such as salivary gland and bone cancers the proportions were 0.2% and 0.8%, respectively. The number of all families was 449,545 and of these 53,132 (11.8%) had concordant cancers.

### 2.2. Risk Estimates for Offspring of Affected Patients and for Siblings

Familial standardized incidence ratio (SIR, also referred to as familial relative risk) for offspring of affected parents was increased for 29/31 cancers; the exceptions being salivary gland and pharyngeal cancers (Table 2). We show data only for combined sexes as none of the sex-specific SIRs were significant. The highest SIR for offspring of affected parents was observed for bone cancer (6.11), followed by small intestinal (5.37) and testicular cancer (4.56). For common familial cancers, the SIRs were 2.05 for prostate, 1.74 for breast and 1.66 for colorectal cancers. For the latter, risk for colon (1.79) was somewhat higher (95%CIs overlapped) than that for rectal cancer (1.61). The highest relative risks between siblings were observed for bone cancer (8.23), Hodgkin disease (6.77), and small intestinal cancer (6.11).

Familial risks between offspring and parents and between siblings may have different causes, as for the latter recessive inheritance and shared childhood environment may play a role. Overall, the relative risk for siblings was higher (SIR 2.07) than it was for offspring and parents (1.86). For individual cancers, a significantly higher risk for siblings than for offspring of affected parents was noted for a total of four cancers. Hodgkin disease was characterized by a large difference (6.77 vs. 2.87), as was stomach cancer (2.74 vs. 1.66). The other sites with a significant difference were the lung and the prostate.

For bone and connective tissue cancer with previously unreported familial risks, we checked SNOMED histology, available in the Swedish Cancer Registry since 1993. For the nine bone cancers in Table 2, four were chondrosarcomas and two were osteosarcomas. Probands of two chondrosarcoma patients were also diagnosed with chondrosarcoma; for other probands, no SNOMED data were available (diagnosis before 1993). Among the 36 connective tissue cancer patients, eight were leiomyosarcomas, five were liposarcomas, and five were sarcomas not otherwise specified; the remaining cases were diverse sarcomas. Only two proband diagnoses matched case diagnoses but most SNOMED data for probands were missing.

### 2.3. Increased Risk When Several Family Members Were Affected

In Table 3, we compare concordant familial risks for those who have one first-degree relative (FDR) diagnosed with cancer to those who have two or more affected relatives. SIRs for persons with one affected FDR are the composite of SIRs from Table 2, with an overall SIR of 1.94. For persons with two or more affected relatives, the overall SIR was 3.33. However, such high-risk, ‘multiplex’ families were found for only 20 cancers (including colorectal cancer). Almost all SIRs for multiplex families were significantly higher than those for in single FDR families. Some SIRs were remarkably high, such as 112.36 for small intestinal and 55.25 for connective tissue cancers. Even for common cancers, such as colon (3.64), lung (3.42), and prostate (3.74) cancers, the SIRs were well above those with a single FDR. However, the total number of cases accounting for the multiplex families was only 4532 of all 69,104 familial cases (6.6%). The proportion for prostate cancer in such high-risk families was 10.5% (2550/24,238), for breast and colorectal cancers, these were 5.5% for both. Prostate cancer accounted for 56.4% of cases in multiplex families, followed by breast (20.1%) and colorectal (9.2%) cancers.

### 2.4. Age-Incidence for Familial and Non-Familial Cancer

We selected 10 common cancers for a detailed study of age-incidence relationships. Familial risk was defined through FDRs. The details of the calculations for these 10 cancers (rate ratios, and 95%CIs) are shown in Appendix A. The first two cancer incidences of the stomach and colorectum are shown in Figure 1. Incidence rates and rate ratios (RRs, familial/non-familial) are shown under each panel. For stomach cancer, an early onset peak reachedRR of 5.2, while in the early middle age it was two and then declined at higher ages. Even for this rare familial cancer, the RRs were significant in almost all five-year age brackets. For colorectal cancer, the highest RR at age 40–44 reached 2.3, followed by a slow decline to 1.3.

For pancreatic cancer, a prominent early age component reached RR of 9.2, while the middle age RR was below three and declined with age (Figure 2). For lung, the RRs peaked at 2.6 at age 40–44, and declined at higher age. Breast cancer showed a modest early onset maximum (2.6) at age 25–29, followed by a steady decline (Figure 3). Endometrial cancer was characterized by an early onset (2.9) and a higher (3.2) early middle age component.

The distributions of RR for prostate and kidney cancers were similar, with the exception that the kidney cancer peak of RR 5.8 was at age 25–29 while the one of RR 6.0 for prostate cancer was at age 40–44 (Figure 4). Furthermore, the shapes of RR distributions for bladder cancer and melanoma showed similarities, RRs reaching 2.4 for bladder cancer and 3.3 for melanoma (Figure 5).

## 3. Discussion

The Swedish FCD is the largest family cancer database in the world, with premium features of nationwide family structures and practically complete data on cancers from the high-quality cancer registry [16,17,18]. The results are not biased by selection of families or inaccurate reporting of cancer in family members. These features guarantee the novelty of the present results: unbiased familial risk estimates and familial proportions even for rare cancers, by the type and number of probands. We discuss below the implications of these findings in terms of germline genetic landscape of familial cancer and the possible clinical impact.

### 3.1. Familial Risk Is a Feature of (All) Cancers

Cancer risks in a 20–84 year old population conferred by affected parents or siblings were about two-fold compared to the risk for individuals with unaffected relatives. The only site with no familial risk was salivary glands presenting only with three familial pairs. In a few sites, the risks were high, five-to-eight-fold, including cancers for which high risks were previously known (small intestinal, testicular, and thyroid cancers and Hodgkin disease) and a novel site (adult bone cancer) [8,9,19]. Furthermore, familial risks for lip and oral cavity, pharyngeal, and connective tissue cancers are novel from the Swedish FCD. Lip, oral cavity, and pharynx have been previously covered as part of upper aerodigestive tract but specific analysis of pharynx did not show significant results [20]. However, positive results have been published from case-control studies and from a cohort study [21,22,23,24,25]. Primary bone and connective tissue cancers present with diverse histologies, but we could not properly test if family members shared histological types of tumors because of missing data from years before 1993. However, two pairs of family members shared chondrosarcoma histology for bone cancer (which also includes tumors of cartilage).

Familial risks were higher for persons whose siblings were affected compared to those whose parents were affected. The difference was significant for a total of four cancers. Such results may indicate recessive genetic effects or deleterious influence of shared environmental risk factors during childhood or adolescence [26]. Recessive effects require mutations in both alleles but monoallelic mutations may also be a risk factor, as reported for the *MUTYH* gene [27]. Panel sequencing has revealed common monoallelic *MUTYH* mutations in colorectal, prostate, and lung cancers [28].

A non-genetic explanation for increased cancer risk is the family member’s concern for his/her own cancer risk when a relative is diagnosed with cancer. We have previously shown that cancer diagnosis in one family member leads to a higher chance of cancer diagnosis in another family member but the increase was limited to the same year. This transient increase was observed for many cancers, including colorectal, lung and prostate cancers; such detection bias was higher for siblings than for offspring of affected parents [29]. In the present study, the increased sibling risk for colon cancer could be related to detection bias. For stomach cancer, a higher concordance within a generation than between generations may be explained by a sharply declining incidence rate [30]. For the high sibling risk of lung cancer, early smoking start among siblings may contribute and facilitate persistent addiction [31]. For prostate cancer, recessive genetic effect cannot be excluded but the above detection bias and uptake of prostate specific antigen testing may preferentially influence brothers [29,32]. For Hodgkin lymphoma, shared childhood social environment has been proposed as a risk factor [33].

We showed in Table 3 that the risks increased by the number of affected relatives. For some common cancers, such as breast, colorectal, and prostate cancers, it has been possible to identify families with many affected relatives, and hence high risks have been shown in earlier studies [2,34]. In the extreme, for prostate cancer risk for a sibling with four affected brothers the cumulative risk was 50%, which would imply dominant inheritance with 100% penetrance [34].

### 3.2. Familial Proportions

Our data provide an answer to the question, ‘how common is familial cancer’. A casual reader may feel that such data are already known. We have been curious about how well this is known and reviewed the 18 chapters on site-specific cancers in the ‘classical’ multi-author treatise,’ The Genetic Basis of Human Cancer’ [35]. Half of the chapters made a reference to family history, but only two specified what was meant [36]. Unqualified statements about family history are commonplace even in expert reviews on specific cancers; unreferenced statements are made ‘X% of the patients have a family history of this cancer’, and such figures circulate in subsequent publications as proven facts [36]. Even if proper citations are given the accuracy of data is not controlled. Many familial risks reported in the literature are based on anecdotal reports on cancers in family members without case confirmation. The accuracy of reporting a family history is notoriously inaccurate for many internal cancers [37,38,39].

Familial proportions are a moving target because they depend on the magnitude of familial risk and population prevalence of each cancer [36]. Thus, it is not surprising that the highest proportions were found for common cancers. Yet, it may be surprising that for prostate cancer familial proportion was as high as 26.4% (every fourth prostate cancer is diagnosed in families of prostate cancer patients), and that for all concordant cancer it was as high as 13.4%.

### 3.3. Genetic Implications

The contribution of germline genetics to the prevalence of cancer is largely cancer specific, ranging from those with prominent high-risk genes on a polygenic background (colorectal and breast cancers and melanoma) to those with a predominant low-risk polygenic background (prostate cancer), and to those with limited contribution by either types of genes (lung and bladder cancer) [40,41,42]. Early onset is a characteristic of heritable cancer, and is also a prominent feature of most familial cancers [43]. Nevertheless, familial clustering is not limited by age and most familial cases may be found at ages when the background incidence is highest [44]. Family studies do not tell the reason for familial clustering but level of risks and detailed age-incidence data may suggest involvement of known germline genetics for the particular cancers.

Among the 10 cancers with detailed age-incidence data, clear early onset risks for stomach, pancreatic, and kidney cancers suggest contribution by known genes. Hereditary stomach cancer is associated with mutations in the *CDH1* gene and it is also manifested in Lynch (hereditary nonpolyposis colorectal cancer) syndrome related to mutations in mismatch repair genes [45,46,47,48]. Next-generation sequencing studies have identified rare germline variants in several other genes, including *BRCA2* and other DNA repair genes [49,50,51]. In kidney cancer, the contribution of von Hippel–Lindau syndrome is a likely explanation for the early onset risk component [52]. In pancreatic cancer, the early onset risk could be due to high risk genes such as *CDKN2A, BRCA1,* and *BRCA2*, or some known rare genes, while the second peak at age 40–49 matches the age of onset of Lynch syndrome [6]. Lynch syndrome is most likely prominently contributing to the main colorectal cancer peak at ages 30–44 years, to the main endometrial cancer peak and to bladder cancer susceptibility [53,54,55]. A recent New York study on bladder (urothelial) cancer found pathogenic variants in 14% of patients, and *MSH2* and *BRCA2* mutations were significantly associated with risk with odds ratios of about four [56]. *CDKN2A* is the main predisposing gene also for melanoma and it contributes to the genetic architecture of melanoma as mutations are found in about 30% of patients in rare families of three or more affected individuals [57]. According to a Swedish study, *CDKN2A* positive families accounted for 11.5% of all melanoma families, and the positivity correlated with the number of affected individuals; in the positive families a median of six melanomas were diagnosed compared to two in mutation negative families [58]. However, the broad age range that we observed suggests that other genes are involved including those associated with skin pigmentation, and hence with possible interactions with ultraviolet radiation [59]. The same kind of genetic background may also apply to breast cancers, for which *BRCA1* and *BRCA2* are prominent predisposing genes for the early onset component but at higher ages other genes are likely to contribute [60].

Lung cancer has the strongest environmental component among these cancers, as shown by spouse correlation [1,2]. In a modeling study, we have estimated, based on the assumptions of heritability of the smoking habit, that shared smoking would contribute to a familial risk of 1.19, which is well below the SIR of 2.14 observed here [61]. However, the problems with such estimates of genetic and environmental effects is that interactions cannot be reliably assessed and nicotine dependence plays an important role in tobacco carcinogenesis [62]. An interesting aspect of the age-incidence relationships for lung cancer was the relatively early peaking and modestly declining risk towards old age, implicating polygenic inheritance and probable interactions with smoking. Genetics of prostate cancer have shown population specific features, particularly related mutations in *HOXB13, NBN,* and *CHEK2;* however mutations in *BRCA2,* mismatch repair genes, *BRCA1*, and *ATM* appear to contribute to risk in many populations [63]. The detected high-risk peak at age 40–44 years could be due to *HOXB13* and other of these genes [63].

### 3.4. Clinical Interventions and Familial Risk

In considering familial cancer, one needs to keep in mind that for certain cancers clinical interventions have probably influenced familial risks in Sweden. For melanoma prevention, national multi-level campaigns were started already in the late 1980s. These targeted physicians, nurses, and the general population [64]. Different population screening approaches were tested and, while being overall beneficial, they also showed that these programs did not reach people in all walks of life [65,66]. Genetic counseling and genetic testing for Lynch syndrome in Sweden was reviewed over a period of 20 years from 1994 to 2014, and 369 families were identified, which the authors assumed to account for one quarter of national Lynch families [67]. Another study reported over 500 Lynch families, and sequenced a cohort of 572 consecutive colorectal cancer patients for mismatch repair gene mutations; pathogenic variants were found in 1.9% of patients, suggesting that universal testing could be equally effective in identifying mutation carriers as the Bethesda criteria [68]. A recent Swedish study on more than 5000 unselected breast cancer patients reported that the prevalence of pathogenic *BRCA1/2* mutations was 1.8%; six in ten *BRCA1/2* mutation carriers were not detected by selective clinical screening of individuals [69]. Only 8% of all patients had been part of a previous clinical screening but their share of the detected mutations was 38%. For prostate cancer, Sweden resisted against prostate specific antigen testing, but once it started in the late 1990s, a huge incidence peak for prostate cancer was observed [32].

How did the above clinical and screening interventions change familial risks? An answer is offered by a comparison of our current study with our earliest comprehensive analysis of familial risk for which cancers were followed-up until 1999; we can assume that, with the exception of melanoma, none of the above interventions played a role for those results [70]. Melanoma risks for offspring with affected parents showed exactly the same SIR of 2.41 then and now. For colorectal and breast cancer the old SIRs were about one decimal unit higher than the present ones, which would be consistent with the younger age of the index population in the old study. For endometrial cancer, there was a larger decrease in the current SIR (old 2.86 to current 1.98), which was also true for prostate cancer (2.71 to 2.08). As for these cancers, familial risk decreased markedly with increasing age (cf. Figure 3 and Figure 4), and it is likely that the age truncation was the main cause for time-dependent decrease in risk. The differences were larger for sibling risks but the siblings were very young (below 62 years) in the old study.

The somewhat surprising conclusion is that the interventions have not influenced familial risks. The main reason may be that the high-risk families account for a small proportion of all familial cancers (6.6% of families had more than two concordant cancers), which was one of the main conclusions of the present study. Another reason is that only a proportion of affected families have been identified as referred to above for Lynch syndrome. Finally, prevention of familial risk is effective only when intervention is successful before cancer diagnosis in yet unaffected carriers. It is not known if preventive counseling of healthy family members in the identified families has been of high priority in Sweden.

### 3.5. Clinical Implications of Familial Risk

The population attributable fraction defines the proportion of a disease that would be prevented if the risk factor or the cause could be avoided, and it is often used in ranking causes of diseases. Familial risk is an important cause of cancer, and it has been ranked as the third most common population attributable fraction after tobacco smoking and unhealthy diet [8]. For known environmental causes of familial clustering, including smoking, alcohol, solar irradiation, infections and other causes, primary prevention or avoidance of risk factors is the most effective method of risk reduction. For genetic causes, however, primary prevention is more difficult and early detection may be the method of choice in catching precursor lesions. Family history has been and still is an important guide in identification of Lynch, breast-ovarian, or other cancer syndromes and in directing clinical management and counseling [3,4,71,72]. However, the power of family history is weakened when penetrance of the genetic effect is low and when families are small. This raises the question about utility of family histories as a guide to genetic background at the time when genetic testing through gene panels and next generation sequencing has entered the oncology routine. Below we discuss some clinical scenarios about management of familial/heritable cancer.

Family history may be redundant if management of cancer patients is based on universal genetic testing with panel (or other) sequencing methods of sufficient sensitivity and specificity [28]. While such approaches are able to find more pathogenic variants than guideline-directed targeted testing, they create a number of problems germane to panel sequencing, which include variants of uncertain significance (VUS), increasing need for counseling and costs [4,7]. VUS increase with increasing panel sizes, and more than one VUS may be detected in a single patient. Counseling is necessary before and after testing, and dealing with VUS is an unsolved problem [4,7]. A practical hindrance to large scale applications of panel sequencing is that the personnel and financial demands would be beyond most health care systems. Offering panel tests also to unaffected family members (cascade testing) would add another dimension to the costs. Even targeted sequencing demands health care resources, and it has been estimated that fewer than 10% of prevalent BRCA1/2 or mismatch repair mutation carriers have been identified in the UK [42,73]. Universal sequencing of cancer patients or of the total population may become more tractable when alternative technologies, such as array-based sequencing methods are being developed [42].

Another scenario, which we proposed based on the present results, is to intensify the application of family and personal history (age of onset and multiple primary cancers, considering also relatives) with separate strategies targeting high-risk families (two or more diagnosed family members and/or known predisposing genes) and low-risk families (one affected family member). The present high-risk (multiplex) families accounted for only 6.6% of all familial cancer cases. Colorectal and breast cancer accounted for 25.5% of these multiplex families, and these and ovarian cancer are the only cancers for which genetic testing has been organized. Predisposing genes for many other high-risk cancers are known, but genetic testing is done only on a case-by-case basis. Thus, there is scope to expand targeted high-risk testing to a larger number of high-risk families considering the most likely predisposing genes, and the likely benefits of gene identification in terms of therapeutic options and preventive counseling of family members. Oncology clinics should continue in managing high-risk families for which genetic testing is justified and counseling of patients and their relatives is required. Moreover, when a mutation test is negative, an appropriate surveillance plan should be developed considering the level of risk in each family. Starting age of screening for unaffected family members should be accordingly recommended [34,74]. In high-risk families multiple primaries are common and these should be considered in counseling. Managing all types of high-risk familial patients and their family members should be built into the clinical care routine with dedicated staff.

For the >90% of familial cancers with a relative risk of about two, genetic testing may not be indicated. However, it is important to be sensitive to rare cancers and small families because a single family member with a rare cancer may signal high risk. All cancer patients are at a risk of second primary cancers and family history is a risk factor for that particular cancer to be diagnosed as second primary cancer, observed for diverse primary cancers, such as breast cancer, melanoma and non-Hodgkin lymphoma [75,76,77]. Thus, patients need counseling about risk factors, surveillance and possible screening tailored to his/her primary cancer. If that cancer may appear as multiple primaries, such as breast or colorectal cancers or melanoma, the risk of second primaries at these sites should be considered in the management plan. Ideally, counseling should be extended to family members but reaching them may be problematic and could perhaps best be conducted through prevention campaigns at primary health care centers [78].

The entrance of large-scale DNA sequencing into oncology clinics may herald a paradigm change towards technology driven patient care as we described above. Powerful technologies often result in hype of promising health benefits which, however, remain to be demonstrated. Our proposed strategy is an extension of the existing practice, aiming at balancing the population burden of familial cancer between rare high-risk individuals with lower population risk. We have to admit that, in spite of recommendations by professional organizations, the evidence for targeted screening of hereditary/familial cancer is not overwhelming. The first and only controlled trial in Lynch syndrome, which was conducted in Finland, showed a significant reduction in subsequent colorectal cancers compared to the control arm of no screening [79]. For ethical reasons, no more such trials have been conducted but results from prospective surveillance have supported the reduction in cancers and deaths [80]. A systematic review on evidence relating to management of *BRCA1/2*-related cancers in women concluded that no single study evaluated the effectiveness of risk assessment, genetic counseling, or genetic testing in reducing incidence and mortality in of *BRCA1/2*-related cancers; a total of 103 studies were evaluated [81]. For melanoma, an Australian expert group found some justification of screening of high-risk groups but little supporting evidence [82]. Whichever strategies are followed in fighting familial cancer, more resources are required. The beneficial population impact should be a fair target for health policy decision makers.

### 3.6. Limitations and Need for Future Studies

The main limitations in this population-based approach are the descriptive nature of the results confined to the population under study. Most causes for familial clustering are unknown, and even though circumstantial evidence suggests involvement of known and unknown genes (or gene-environment interactions), their role remains speculative. We lack information on potential confounders and causes of familial clustering, including lifestyle factors. However, adjustment for socio-economic factors helps mitigate their influence [83]. In Discussion, we speculate about the predisposing genes that may contribute to the observed familial risks; we included only some of the most common predisposing genes as for rarer genes it would have been necessary to carry out detailed analysis, including discordant cancers and histologies which were beyond the present scope. Another limitation is that the population under study is not ‘fully aged’, i.e., the 20–84 year old index population is a segment of the total population, and the median age of cancers in the index population was some 10 years below that in the Swedish Cancer Registry [84]. This limited the use of children of the index persons as FDRs. Thus, true multigenerational studies are not possible in the current FCD, lacking information on past generations, which are available in population databases in Iceland and Utah [85,86,87]. In future studies, it will be possible to test more complex and extended family structures but this does not change the truth that genetic sharing is halved in every added generation. Thus, analysis by FDR is most powerful in showing genetic effects. Our experience from using a much smaller FCD up to 20 years ago, as discussed above, showed that the magnitude of familial risk was only marginally changed, dependent on age of the population. Yet, the benefit of the current expanded FCD was the statistical power of finding familial associations in rare cancers [70,88].

## 4. Conclusions

Next-generation sequencing methods have made germline analysis of cancer more tangible but also revealed complexities and new dilemmas. Pathogenic or likely pathogenic mutations in *BRCA1/2*, mismatch repair and other cancer predisposing genes are found in many cancers for which genetic association studies are not available, leaving the causal role of mutations open. The familial landscape of cancer, which we defined through this largest family study yet published, demarcated a familial proportion of 13.2% of all cancers, considering concordant cancer in FDR; 6.6% of cancer families (0.9% of all cancers) included at three discordant cancer patients. Among this 6.6%, systematic genetic testing is available only for a quarter of patients, and this share should be expanded based on family and personal histories. For the major proportion of familial clustering, genes remain unknown or include low-risk genes with possible environmental interactions; some familial clustering may be due to pure non-genetic factors including diet, tobacco and alcohol use and environmental carcinogens such as pollution or radon. Nevertheless, medical and behavioral intervention may be indicated, including screening recommendations and avoidance of carcinogenic exposure. The readily available information of family history deserves more attention from first diagnosis through referral to the oncology clinic, enabling establishment of clinical counseling strategies, individually tailored to screening and prevention needs of patients and their family members.

## 5. Methods

### 5.1. Data Source and Patients

The FCD was used to define the index population (second generation or offspring generation) born after 1931 and the parental generation who were parents of the index population. These two generations were used to estimate familial cancer risks for the 31 most common cancer sites and, in addition, colorectal cancer. The diagnostic age was defined to over 19 years in order to exclude childhood cancers with divergent etiologies compared to adult cancers [89]. FCD comprises information collected at the Center for Primary Health Care Research, Lund University, Malmö, from nationwide databases of the Multigeneration Register, censuses and death notifications maintained at Statistics Sweden, and information from the Swedish Cancer Registry. In its latest update from 2016, FCD includes 16.8 million individuals, where all people born in Sweden from 1932 onwards (the offspring generation) were linked to their biological parents (the parental generation). Register linkages were performed at the individual level via the national 10-digit civic registration number. In the linked dataset, civic registration numbers were replaced with serial numbers to ensure anonymity.

A total of 2,013,623 medically verified cancers were recorded in the Cancer Registry from 1958 to 2016. The 7th revision of the International Classification of Diseases (ICD-7) and later revisions were used to identify the cancer type. SNOMED histology was used in some analyses. In order to investigate cancer risks for offspring of affected parents as well as risk shared by siblings, all individuals with identified parents were selected for the analysis, totaling in 9,338,882 index individuals. Among them, 552,953 were diagnosed with one of the cancer sites under study.

### 5.2. Relative Risk Estimation

Familial risk was assessed for the offspring generation by estimating standardized incidence ratios (SIRs) as the ratio of observed (O) to expected number of cases. The observed numbers were cancers in the 20–84 year old index populations, whose FDRs were diagnosed with cancer; in the case of siblings, the observed number included all affected siblings. The expected numbers were calculated for all individuals without cancer in family members (i.e., the Swedish population minus familial cases), and the rates were standardized by 5-year-age, gender, period (5 years group), latest educational attainment (as proxy socioeconomic status), and residential area. As the SIR calculation was based on person-years at risk, it was independent of family size. The 95% confidence interval (95%CI) of the SIR was calculated assuming a Poisson distribution.

### 5.3. Familial Proportion, Different Levels of Family History, and Age-Incidence Ratios

Familial proportion was defined in the index population as the percentage of concordant familial cancers (including offspring of affected parents and siblings) of all cancer. Cancers in parents contributed only to the definition of family history but not to case numbers.

Independent groups for familial relationships were considered in order to examine the differences in the familial risk shared by parents and siblings, and the risk shared by multiple affected relatives. The dependence of familial risk on diagnostic age (age-incidence) was investigated considering all first-degree relatives (FDRs) as probands for 10 common cancers. Plots of familial and non-familial incidence rates in 5-year intervals were generated. RRs between familial and non-familial rates were given and 95%CIs were calculated for each age band and for any age.

### 5.4. Calling Significant Difference

The difference between two SIRs was called significant when their 95%CIs did not overlap. The word ‘significance’ was not always repeated but a reference to a difference implied that it was significant, unless the opposite was stated.

## Figures and Tables

**Figure 1 cancers-13-04385-f001:**
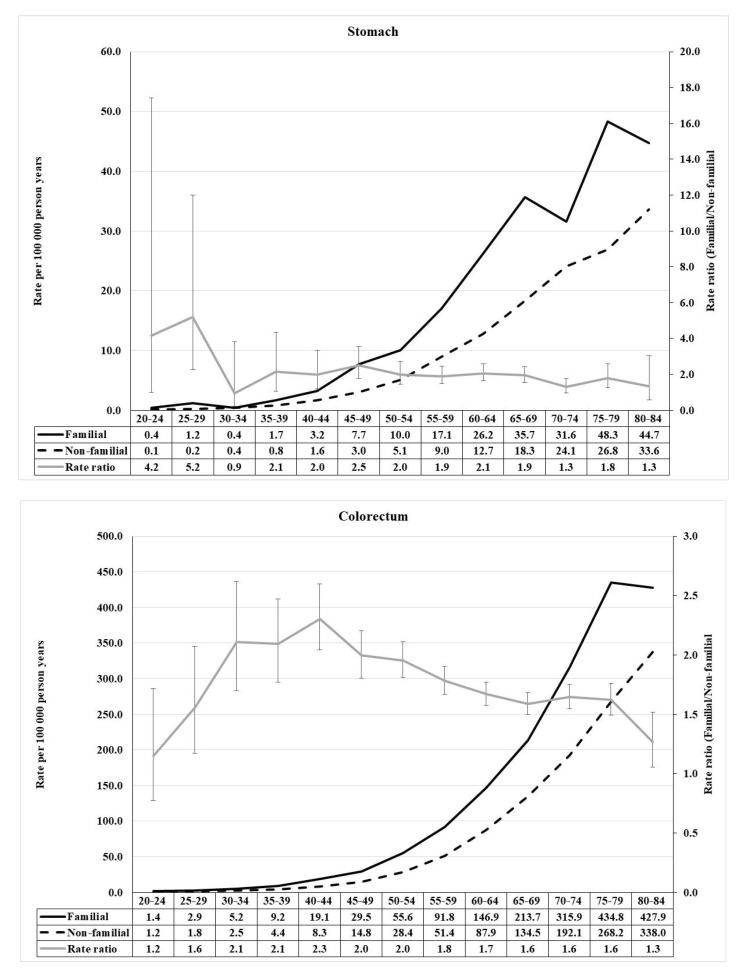
Age-specific incidence rate and rate ratio of stomach (**upper**) and colorectal (**lower**) cancer in population with and without family history of concordant cancer.

**Figure 2 cancers-13-04385-f002:**
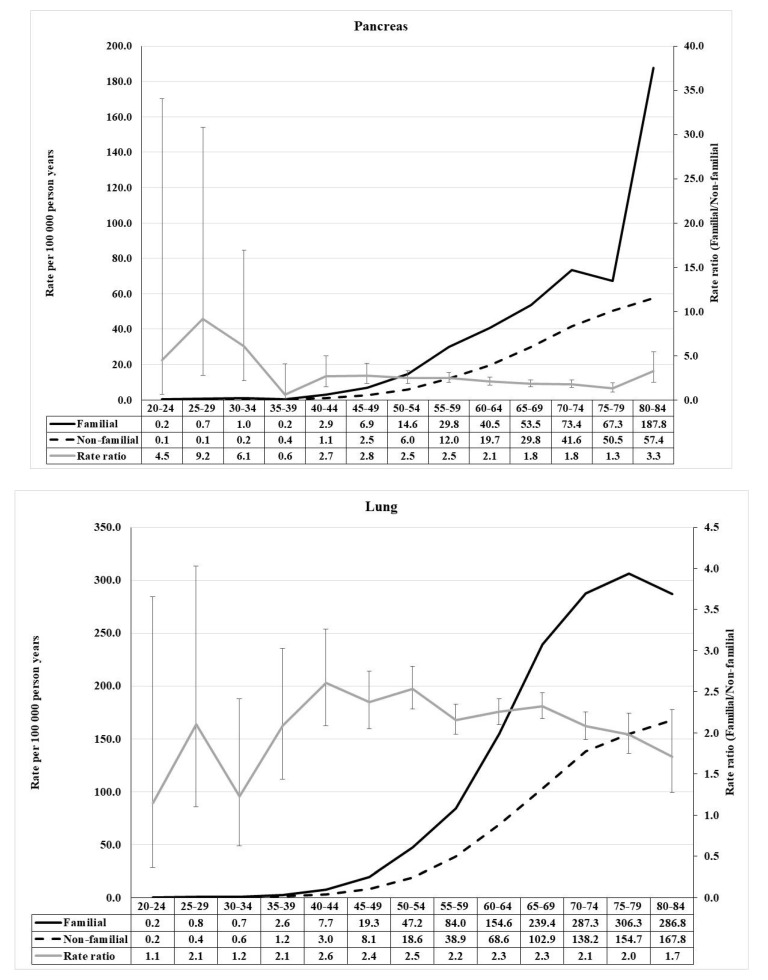
Age-specific incidence rate and rate ratio of pancreas (**upper**) and lung (**lower**) cancer in population with and without family history of concordant cancer.

**Figure 3 cancers-13-04385-f003:**
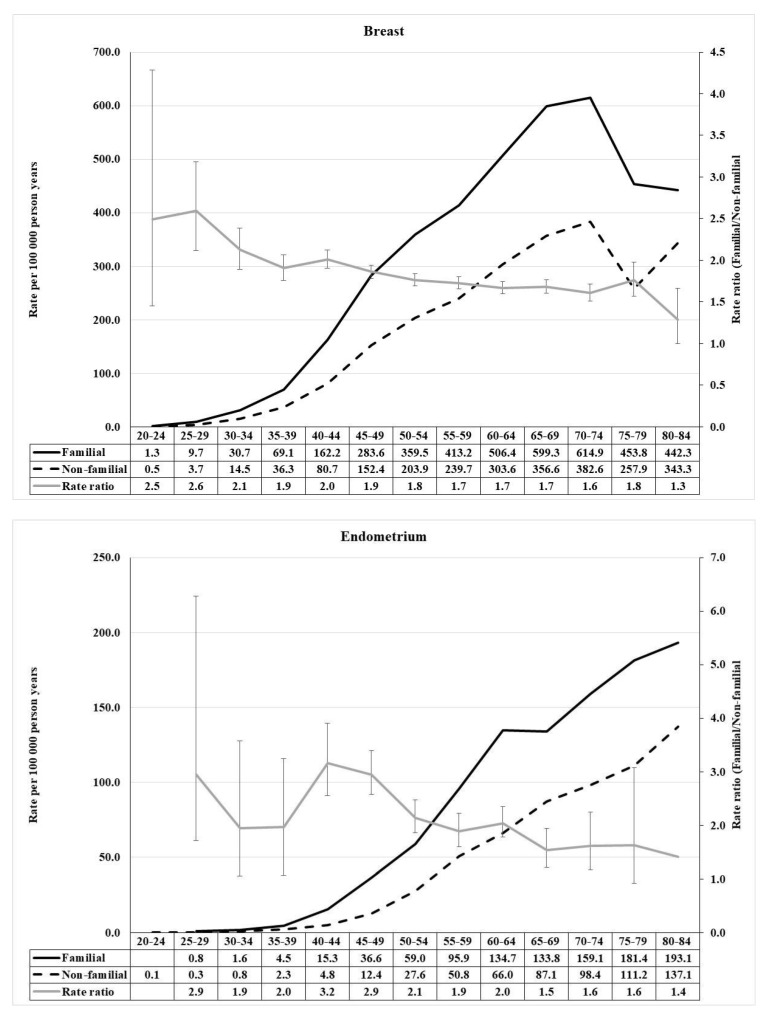
Age-specific incidence rate and rate ratio of female breast (**upper**) and endometrium (**lower**) cancer in population with and without family history of concordant cancer.

**Figure 4 cancers-13-04385-f004:**
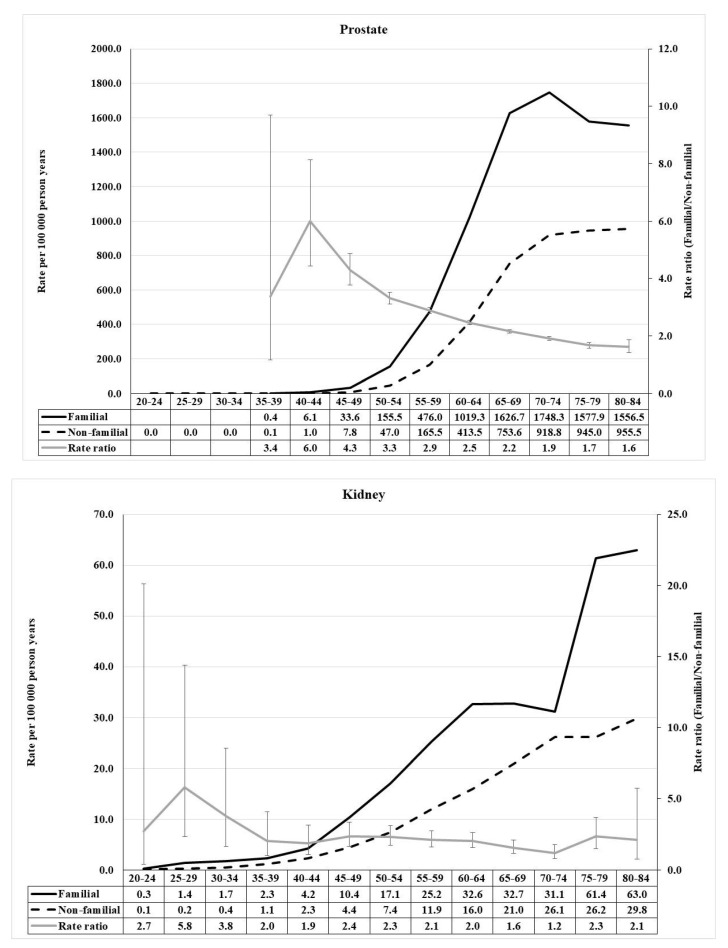
Age-specific incidence rate and rate ratio of prostate (**upper**) and kidney (renal parenchyma) (**lower**) cancer in population with and without family history of concordant cancer.

**Figure 5 cancers-13-04385-f005:**
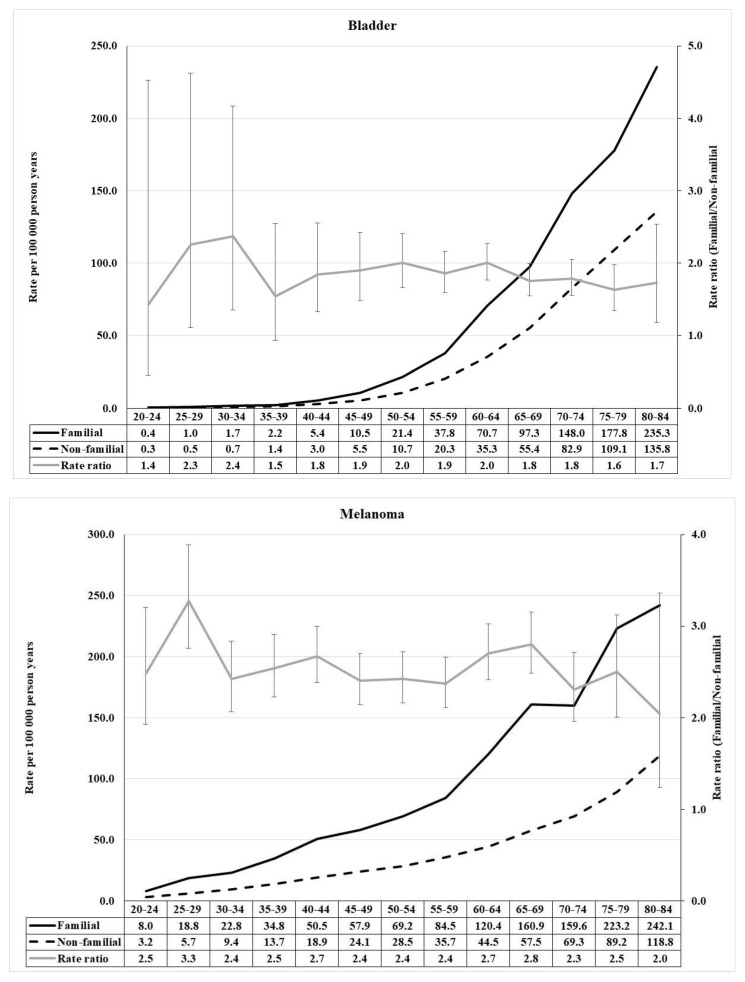
Age-specific incidence rate and rate ratio of bladder (urinary bladder) cancer (**upper**) and melanoma (**lower**) in population with and without family history of concordant cancer.

**Table 1 cancers-13-04385-t001:** Number of cancers, median ages, incidence rates, and number of familial cancer in the 20 to 84 year old index population (N = 9,338,882) during 1958–2016.

Cancer Site	ICD-7 Codes	Total Numbers *		Median Age	Incidence Rate per 100,000 Person Years			Family History of Concordant Cancer	
		No.	%		IR	95% CI		No.	Familial Proportion %
Lip, oral cavity	140, 141, 143, 144	5772	1.1	60	1.86	1.81	1.91	119	2.1
Salivary glands	142	1299	0.2	53	0.42	0.40	0.44	3	0.2
Pharynx	145, 146, 147, 148	3718	0.7	58	1.20	1.16	1.24	45	1.2
Esophagus	150	3854	0.7	63	1.24	1.20	1.28	73	1.9
Stomach	151	6980	1.3	61	2.25	2.20	2.30	411	5.9
Small intestine	152	2442	0.5	60	0.79	0.76	0.82	54	2.2
Colorectum	153, 154, excluding anus	48,803	9.3	63	15.72	15.58	15.86	7650	15.7
Colon	153	30,819	5.9	64	9.93	9.82	10.04	3462	11.2
Rectum	154, excluding anus	17,984	3.4	62	5.79	5.71	5.88	1054	5.9
Liver, primary	155.0	4113	0.8	63	1.32	1.28	1.37	86	2.1
Pancreas	157	9557	1.8	64	3.08	3.02	3.14	465	4.9
Lung	162, 163	33,121	6.3	64	10.67	10.55	10.78	4291	13.0
Breast, female	170	95,756	18.2	55	30.85	30.65	31.04	16,718	17.5
Cervix	171	10,542	2.0	40	6.94	6.81	7.08	223	2.1
Endometrium	172, 174	16,012	3.0	61	10.55	10.38	10.71	821	5.1
Ovary	175	11,554	2.2	55	7.61	7.47	7.75	495	4.3
Prostate	177	91,696	17.5	66	57.82	57.44	58.19	24,238	26.4
Testis	178	8272	1.6	33	5.22	5.10	5.33	158	1.9
Kidney parenchyma	180.0	8574	1.6	60	2.76	2.70	2.82	325	3.8
Bladder	181.0	18,738	3.6	64	6.04	5.95	6.13	1319	7.0
Melanoma	190	37,842	7.2	52	12.19	12.07	12.31	2715	7.2
Skin, squamous cell	191	19,014	3.6	67	6.12	6.04	6.21	1476	7.8
Eye	192	1607	0.3	57	0.52	0.49	0.54	10	0.6
Nervous system	193	21,560	4.1	51	6.95	6.85	7.04	790	3.7
Thyroid gland, adenocarcinoma	194 + PAD 096	6044	1.2	42	1.95	1.90	2.00	136	2.3
Endocrine glands	195	11,713	2.2	53	3.77	3.70	3.84	341	2.9
Bone	196	1186	0.2	39	0.38	0.36	0.40	9	0.8
Connective tissue	197	3754	0.7	52	1.21	1.17	1.25	34	0.9
Hodgkin disease	201	4064	0.8	32	1.31	1.27	1.35	57	1.4
Non-Hodgkin lymphoma	200, 202	17,197	3.3	59	5.54	5.46	5.62	743	4.3
Myeloma	203	5618	1.1	63	1.81	1.76	1.86	140	2.5
Leukemia	204–209	14,617	2.8	60	4.71	4.63	4.78	643	4.4
All above		525,019	100.0	60	169.12	168.66	169.58	69,104	13.2

*: Total number of cancers was 552,953 including 27,934 diverse rare cancers, which were not included in this study. IR: Incidence rate per 100,000 person years; PAD: Pathological anatomical diagnosis.

**Table 2 cancers-13-04385-t002:** Familial risk for index persons whose parents or siblings were diagnosed with concordant cancer.

Cancer Site	Only Parent Diagnosed with Concordant Cancer				Only Sibling Diagnosed with Concordant Cancer			
	O	SIR	95% CI		O	SIR	95% CI	
Lip	73	**1.78**	**1.39**	**2.24**	46	**2.44**	**1.79**	**3.26**
Salivary gland	3	1.80	0.34	5.34				
Pharynx	15	1.66	0.93	2.75	30	**3.88**	**2.62**	**5.55**
Esophagus	43	**2.37**	**1.71**	**3.19**	30	**2.91**	**1.96**	**4.16**
Stomach *	307	**1.66**	**1.48**	**1.85**	94	**2.74**	**2.22**	**3.36**
Small intestine	31	**5.37**	**3.65**	**7.63**	21	**6.11**	**3.78**	**9.36**
Colorectum	4981	**1.66**	**1.62**	**1.71**	2252	**1.76**	**1.69**	**1.84**
Colon	2249	**1.79**	**1.71**	**1.86**	1060	**1.96**	**1.84**	**2.08**
Rectum	696	**1.61**	**1.49**	**1.73**	341	**1.69**	**1.52**	**1.88**
Liver, primary	56	**2.30**	**1.73**	**2.98**	28	**2.65**	**1.76**	**3.84**
Pancreas	320	**1.99**	**1.78**	**2.22**	135	**2.17**	**1.82**	**2.57**
Lung *	2247	**1.86**	**1.79**	**1.94**	1875	**2.50**	**2.39**	**2.62**
Breast	9154	**1.74**	**1.70**	**1.77**	6573	**1.73**	**1.69**	**1.77**
Cervix	155	**1.51**	**1.28**	**1.76**	68	**1.58**	**1.23**	**2.01**
Endometrium	511	**1.94**	**1.78**	**2.12**	288	**1.82**	**1.61**	**2.04**
Ovary	319	**2.32**	**2.08**	**2.59**	162	**2.31**	**1.97**	**2.70**
Prostate *	11,791	**2.05**	**2.01**	**2.09**	9897	**2.38**	**2.33**	**2.43**
Testis	51	**4.56**	**3.39**	**5.99**	107	**5.56**	**4.55**	**6.72**
Kidney	211	**1.77**	**1.54**	**2.03**	108	**2.38**	**1.95**	**2.87**
Bladder	835	**1.76**	**1.64**	**1.88**	459	**1.94**	**1.77**	**2.12**
Melanoma	1360	**2.37**	**2.24**	**2.49**	1260	**2.45**	**2.32**	**2.59**
Skin	991	**1.98**	**1.86**	**2.11**	426	**1.89**	**1.72**	**2.08**
Eye	8	**3.30**	**1.41**	**6.53**	2	1.42	0.13	5.24
Nervous system	443	**1.49**	**1.36**	**1.64**	320	**1.64**	**1.47**	**1.83**
Thyroid gland, adenocarcinoma	71	**3.78**	**2.95**	**4.77**	65	**4.98**	**3.84**	**6.35**
Endocrine glands	201	**2.03**	**1.76**	**2.34**	124	**2.00**	**1.67**	**2.39**
Bone	5	**6.11**	**1.93**	**14.36**	4	**8.23**	**2.14**	**21.29**
Connective tissue	22	**1.76**	**1.10**	**2.67**	10	1.61	0.77	2.97
Hodgkin disease *	23	**2.87**	**1.81**	**4.31**	34	**6.77**	**4.69**	**9.47**
Non-Hodgkin lymphoma	443	**1.66**	**1.51**	**1.82**	294	**1.80**	**1.60**	**2.02**
Myeloma	96	**1.92**	**1.56**	**2.35**	44	**2.00**	**1.45**	**2.69**
Leukemia	409	**1.83**	**1.66**	**2.02**	217	**1.97**	**1.72**	**2.26**
All above *	38,120	**1.86**	**1.84**	**1.88**	26,374	**2.07**	**2.04**	**2.09**

O = Observed; SIR = Standardized incidence ratio; CI = Confidence intervals. *: SIRs between cases with parental and sibling probands were significant (non-overlapping 95%CIs).

**Table 3 cancers-13-04385-t003:** Concordant familial risks when one or at least two probands were diagnosed with cancer.

Cancer Site	One Family Member Diagnosed with Concordant Cancer				Two or More Family Members Diagnosed with Concordant Cancer			
	O	SIR	95% CI		O	SIR	95% CI	
Lip	119	**2.00**	**1.66**	**2.40**				
Salivary gland	3	1.19	0.22	3.51				
Pharynx	45	**2.70**	**1.97**	**3.62**				
Esophagus	73	**2.59**	**2.03**	**3.25**				
Stomach	401	**1.83**	**1.65**	**2.01**	10	**5.55**	**2.64**	**10.25**
Small intestine	52	**5.64**	**4.21**	**7.40**	2	**112.36**	**10.59**	**413.21**
Colorectum	7233	**1.70**	**1.66**	**1.74**	417	**2.76**	**2.51**	**3.04**
Colon	3309	**1.84**	**1.78**	**1.90**	153	**3.64**	**3.08**	**4.26**
Rectum	1037	**1.64**	**1.54**	**1.74**	17	**2.20**	**1.28**	**3.52**
Liver, primary	84	**2.40**	**1.92**	**2.98**	2	**12.57**	**1.19**	**46.23**
Pancreas	455	**2.04**	**1.86**	**2.24**	10	**4.96**	**2.36**	**9.15**
Lung	4122	**2.11**	**2.05**	**2.17**	169	**3.42**	**2.93**	**3.98**
Breast	15,805	**1.74**	**1.71**	**1.76**	913	**2.50**	**2.34**	**2.67**
Cervix	223	**1.54**	**1.35**	**1.76**				
Endometrium	799	**1.90**	**1.77**	**2.03**	22	**5.58**	**3.49**	**8.46**
Ovary	481	**2.32**	**2.12**	**2.54**	14	**8.99**	**4.90**	**15.13**
Prostate	21,688	**2.20**	**2.17**	**2.23**	2550	**3.74**	**3.60**	**3.89**
Testis	158	**5.24**	**4.45**	**6.12**				
Kidney parenchyma	319	**1.94**	**1.73**	**2.16**	6	**5.17**	**1.86**	**11.33**
Bladder	1294	**1.82**	**1.72**	**1.92**	25	**2.48**	**1.60**	**3.67**
Melanoma	2620	**2.41**	**2.32**	**2.50**	95	**5.68**	**4.59**	**6.94**
Skin	1417	**1.96**	**1.86**	**2.06**	59	**4.60**	**3.50**	**5.94**
Eye	10	**2.62**	**1.25**	**4.83**				
Nervous system	763	**1.55**	**1.44**	**1.67**	27	**6.24**	**4.11**	**9.09**
Thyroid gland, adenomcarcinoma	136	**4.33**	**3.63**	**5.12**				
Endocrine glands	325	**2.02**	**1.81**	**2.26**	16	**14.82**	**8.45**	**24.11**
Bone	9	**6.93**	**3.14**	**13.20**				
Connective tissue	32	**1.71**	**1.17**	**2.42**	2	**55.25**	**5.21**	**203.18**
Hodgkin disease	57	**4.40**	**3.33**	**5.71**				
Non-Hodgkin lymphoma	737	**1.71**	**1.59**	**1.84**	6	1.51	0.54	3.31
Myeloma	140	**1.97**	**1.66**	**2.32**				
Leukemia	626	**1.88**	**1.73**	**2.03**	17	**5.03**	**2.92**	**8.06**
All above	64,572	**1.94**	**1.93**	**1.96**	4532	**3.33**	**3.24**	**3.43**

O = Observed; SIR = Standardized incidence ratio; CI = Confidence intervals.

## Data Availability

Data used were issued by the National Board of Health and Welfare, Stockholm, to Kristina and Jan Sundquist for exclusive institutional use. Any data requests should be directed to the National Board of Health and Welfare, Stockholm.

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
