# Peer review of "Familial Risks and Proportions Describing Population Landscape of Familial Cancer"

_cancers, 2021, doi:10.3390/cancers13174385_

Round 1

Reviewer 1 Report

In the manuscript entitled “Familial Risks and Proportion Describing Population Landscape of Familial Cancer”, Hemmminki K. et al., retrieved data from 31 most common cancers from the Swedish Family-Cancer Database. Moreover the authors estimated the familial relative risks between persons with or without family history of the same cancer in first-degree relatives.  

Overall, the findings provided are not always clearly shown and / or sufficiently discussed and there are some points that should be addressed. Moreover, the English in the present manuscript require an editing of English language, style and major improvements.

Major Comments:

Simple Summary:

Page 1 (lines 29-32): The authors stated that “The aim of the present study was to highlight the full scope of familial cancers were obtained from the Swedish Family-Cancer Database and familial relative risks (SIRs) were estimated between persons with or without family history of the same cancer in first-degree relatives”.

The aim of the study is not very clear, the authors should better describe the aim of the study in this part of the text. Moreover, the abbreviation “SIRs” has been associated with different explanations: i) familial relative risks (SIRs); ii) standardized incidence ratio (SIRs). Have “familial relative risks” and “standardized incidence ratio” the same significance? The authors should uniformly report the SIRs explanation in all part of the main text.

Introduction:

Page 2 (lines 86-88): The authors stated that “In the present article we aim at describing a nationwide landscape of concordant familial cancer by defining the risk by the proband (parent or sibling), number of affected probands and, for common cancer, by exact diagnostic age of familial cases”. The authors should better describe the aim of the study in this part of the text. Moreover the English of this phrase should be improved.

Methods:

Page 16 (lines 467- 516): this section of the manuscript should be moved after the “Introduction” and before the “Results” Section.

Results:

Page 6; Paragraph 2.4 “Age-incidence for familial and non-familial cancer”

The authors should indicate in the “Material and Methods” section the criteria used to define the “Age-incidence”.

Moreover, the authors should indicate the criteria used to define the non-familial cases.

Another aspect that should be better described regards the “Rate ratio”. The rate ratio (RR) calculation should be reported in material and methods section. Moreover, in the text this ratio has been reported also as relative risks (RR). The authors should uniformly report the “RR” explanation in all part of the main text.

Discussion:

The discussion of the presented manuscript comprise six subparagraph and is composed by five pages. Overall, the discussion is too vast and far exceed the argued results presented in the main manuscript. For these reasons this part of the text is difficult to follow in relation to the results reported in the main text. Therefore, it is suggest to reduce the length of this part of the text, better focusing the discussion on the results to be argued.

Author Response

We are grateful to the expert reviewer for his comments. 

Overall, the findings provided are not always clearly shown and / or sufficiently discussed and there are some points that should be addressed. Moreover, the English in the present manuscript require an editing of English language, style and major improvements.

Major Comments:

Simple Summary:

Page 1 (lines 29-32): The authors stated that “The aim of the present study was to highlight the full scope of familial cancers were obtained from the Swedish Family-Cancer Database and familial relative risks (SIRs) were estimated between persons with or without family history of the same cancer in first-degree relatives”.

The aim of the study is not very clear, the authors should better describe the aim of the study in this part of the text. Moreover, the abbreviation “SIRs” has been associated with different explanations: i) familial relative risks (SIRs); ii) standardized incidence ratio (SIRs). Have “familial relative risks” and “standardized incidence ratio” the same significance? The authors should uniformly report the SIRs explanation in all part of the main text.

 Aims were modified. Familial risk was defined in the first sentence in Results 2.2.

Introduction:

Page 2 (lines 86-88): The authors stated that “In the present article we aim at describing a nationwide landscape of concordant familial cancer by defining the risk by the proband (parent or sibling), number of affected probands and, for common cancer, by exact diagnostic age of familial cases”. The authors should better describe the aim of the study in this part of the text. Moreover the English of this phrase should be improved.

 Aims were modified.

English has been edited by Patrick Reilly. 

Methods:

Page 16 (lines 467- 516): this section of the manuscript should be moved after the “Introduction” and before the “Results” Section.

We have followed the journal instructions. 

Results:

Page 6; Paragraph 2.4 “Age-incidence for familial and non-familial cancer”

The authors should indicate in the “Material and Methods” section the criteria used to define the “Age-incidence”.

Specified in Methods section 4.3.

Moreover, the authors should indicate the criteria used to define the non-familial cases.

This is given in section 4.2 of methods: The expected numbers were calculated for all individuals without cancer in family members (i.e., the Swedish population minus familial cases)…

Another aspect that should be better described regards the “Rate ratio”. The rate ratio (RR) calculation should be reported in material and methods section. Moreover, in the text this ratio has been reported also as relative risks (RR). The authors should uniformly report the “RR” explanation in all part of the main text.

Terms were unified.

Discussion:

The discussion of the presented manuscript comprise six subparagraph and is composed by five pages. Overall, the discussion is too vast and far exceed the argued results presented in the main manuscript. For these reasons this part of the text is difficult to follow in relation to the results reported in the main text. Therefore, it is suggest to reduce the length of this part of the text, better focusing the discussion on the results to be argued.

We agree with the reviewer that for a standard article the Discussion is long. However, considering that the article is intended for a thematic issue on familial cancer, and we have covered ALL cancers with many clinical implications at the time of related paradigm changes we think that the discussion should not be shortened.

Reviewer 2 Report

Authors should be congratulated by the initiative to analyse the whole database for the familial cancer risk in Sweden.

As a suggestion authors are also encouraged to study  rare inherited cancers, derived from the mutations of known genes, like familial colorectal polyposis cancer, von Hippel Lindau, Paraganglioma and Pheochromocytoma  cancers derived from SDH mutations, etc This study could be a continuation for rare  hereditary cancers with more specific conclusions of  genetic screening for target genes.

In general the manuscript is well written, well presented. It is suitable for publication but  this reviewer feels some comments are important to be answered:

  1. Introduction goes directly to the main issue of the paper which is good. However, A paragraph commenting hereditary cancers, like the von Hippel Lindau syndrome, and the paraganglioma and pheochromocytoma tumors derived from mutations in VHL and SDH genes, should be included, just at the beginning of the introduction after the definition of Hereditary cancer.
  2. In the Results, page 5 at the 2.3 section,  overall SIRs of 1.94 in Table 2 and 3.33 in Table 3 are mentioned. Please, explain the meaning of the overall SIR, and how this figure was obtained.
  3. A small typo is found in page 6 at the first paragraph before the Table 3. Prostate cancer accounted for 56.4% of cases in multiplex families, please, correct followed (instead of follow).
  4. In section 3.1 before the reference number 20, I think there is a little typo:“analysis of pharynx did not shown” , i.e, the n should be deleted.
  5. In the same section and paragraph, after references 21-25. Bone and connective…….tissue cancers. It should be stated that bone cancer is primary and metastases from other primary tumors were discarded, if this is the case. In case it is not sure the statement should be included equally.
  6. Page 8 at the end of the paragraph before the section 3.2. The last sentence mentions a sibling with 4 affected brothers and a cumulative risk of 50%, implying a dominant inheritance with 100% penetrance. Could you mention, or figure out which is the involved gene mutated in this case?
  7. In section 3.3 Genetic implications, again the same observation mentioned before: the von Hippel Lindau syndrome with the hemangioblastomas in CNS, kidney, and suprarenal glands should be included associated to mutations in VHL gene. And the paraganglioma and pheochromocytoma syndrome with the SDHA, B, C mutated genes should be also added.

Author Response

We appreciated the positive comments of Reviewer 2.

Authors should be congratulated by the initiative to analyse the whole database for the familial cancer risk in Sweden.

As a suggestion authors are also encouraged to study  rare inherited cancers, derived from the mutations of known genes, like familial colorectal polyposis cancer, von Hippel Lindau, Paraganglioma and Pheochromocytoma  cancers derived from SDH mutations, etc This study could be a continuation for rare  hereditary cancers with more specific conclusions of  genetic screening for target genes.

We sympathize with the comment but as we are limited to concordant cancer we cannot target rare cancer syndromes. This is pointed out in Discussion under limitations 3.6. 

In general the manuscript is well written, well presented. It is suitable for publication but  this reviewer feels some comments are important to be answered:

  1. Introduction goes directly to the main issue of the paper which is good. However, A paragraph commenting hereditary cancers, like the von Hippel Lindau syndrome, and the paraganglioma and pheochromocytoma tumors derived from mutations in VHL and SDH genes, should be included, just at the beginning of the introduction after the definition of Hereditary cancer.

See above.

  1. In the Results, page 5 at the 2.3 section, overall SIRs of 1.94 in Table 2 and 3.33 in Table 3 are mentioned. Please, explain the meaning of the overall SIR, and how this figure was obtained.
  2. A small typo is found in page 6 at the first paragraph before the Table 3. Prostate cancer accounted for 56.4% of cases in multiplex families, please, correct followed (instead of follow).
  3. In section 3.1 before the reference number 20, I think there is a little typo:“analysis of pharynx did not shown” , i.e, the n should be deleted.
  4. In the same section and paragraph, after references 21-25. Bone and connective…….tissue cancers. It should be stated that bone cancer is primary and metastases from other primary tumors were discarded, if this is the case. In case it is not sure the statement should be included equally.

Points 2 to 5 were modified as suggested.

  1. Page 8 at the end of the paragraph before the section 3.2. The last sentence mentions a sibling with 4 affected brothers and a cumulative risk of 50%, implying a dominant inheritance with 100% penetrance. Could you mention, or figure out which is the involved gene mutated in this case?

This is already covered at the end of Discussion paragraph 3.3.

  1. In section 3.3 Genetic implications, again the same observation mentioned before: the von Hippel Lindau syndrome with the hemangioblastomas in CNS, kidney, and suprarenal glands should be included associated to mutations in VHL gene. And the paraganglioma and pheochromocytoma syndrome with the SDHA, B, C mutated genes should be also added.

We refer to the first comment and Discussion paragraph 3.3 which goes as far as we can judge from concordant associations.